# Research on an Enhanced Detuned-Loading Effect in Integrated Two-Section DFB Lasers with High Modulation Bandwidths

**DOI:** 10.3390/mi14111994

**Published:** 2023-10-27

**Authors:** Yunshan Zhang, Hongming Gu, Guolong Ma, Shijian Guan, Tao Fang, Xiangfei Chen

**Affiliations:** 1College of Electronic and Optical Engineering and College of Flexible Electronics (Future Technology), Nanjing University of Posts and Telecommunications, Nanjing 210023, China; yszhang@njupt.edu.cn (Y.Z.); 1021020722@njupt.edu.cn (H.G.); 18066062517@163.com (G.M.); 2College of Engineering and Applied Sciences, Nanjing University, Nanjing 210023, China; dg21340002@smail.nju.edu.cn (S.G.); fangt@nju.edu.cn (T.F.)

**Keywords:** distributed-feedback (DFB) semiconductor lasers, modulation bandwidth enhancement, detuned-loading effect, reconstruction-equivalent-chirp (REC) technique

## Abstract

A novel high-speed directly modulated two-section distributed-feedback (TS-DFB) semiconductor laser based on the detuned-loading effect is proposed and simulated. A grating structure is designed by the reconstruction-equivalent-chirp (REC) technique. A π phase shift is introduced into the reflection grating, which can provide a narrow-band reflection region with a sharp falling slope on both sides of the reflection spectrum, thus enhancing the detuned-loading effect. Owing to its unique dual-falling-edges structure, the bandwidth can be improved even when the lasing wavelength shifts beyond the left falling edge due to a thermal effect in the actual test, in which condition the detuned-loading effect can be used twice, which greatly improves the yield. The modulation bandwidth is increased from 17.5 GHz for a single DFB laser to around 24 GHz when the lasing wavelength is located on the left falling edge of the TS-DFB laser based on the detuned-loading effect, and it can be increased to 22 GHz for the right side. An eight-channel laser array with precise wavelength spacing is investigated, with a side-mode suppression ratio (SMSR) >36 dB. In addition, TS-DFB lasers with uniform reflection gratings are studied, and simulated results show that the modulation characteristic is far inferior to the laser with a phase-shifted grating reflector.

## 1. Introduction

With the explosive increasing demand for network traffic, a reliable high-speed light source enabling high-capacity optical network communication is urgently needed. The directly modulated semiconductor laser (DML) is considered as an ideal light source and has been widely applied in optical data transmission systems for its high energy efficiency, low cost and compact size.

Relaxation oscillation frequency is the main factor limiting the transfer rate and capacity of DMLs. To increase the relaxation oscillation frequency, shortening the cavity length [1,2,3] and the buried heterostructure (BH) [4,5,6] are always applied in the laser structure. Buried heterostructures can reduce the leakage current, restrict the current and light field, and have better threshold characteristics and slope efficiency. Shortening the cavity length can reduce the volume of the active region and improve the modulation rate of the laser. However, less feedback in the short cavity will lead to higher threshold gain, and the complex fabrication process of buried heterostructures will greatly increase the cost.

Many other methods have also been proposed to increase the modulation bandwidth, such as the detuned-loading effect [7,8,9,10,11], optical injection locking [12,13,14] and photon–photon resonance [15,16,17,18]. However, most of these schemes need to use multi-segment-structure lasers, so the butt-joint growth process is unavoidable and the integration between the active and passive region greatly increases the difficulty and cost of manufacture.

In this paper, we demonstrate a novel high-speed directly modulated two-section distributed-feedback (TS-DFB) semiconductor laser based on the detuned-loading effect. The TS-DFB laser is designed by the reconstruction-equivalent-chirp (REC) technique [19,20]. Using the REC technique, the production standard can be reduced to the micron level, which would greatly reduce the manufacturing cost. This device consists of two sections: a DFB section and a grating reflector (GR) section. These two sections share the same wafer structure, avoiding the complex butt-joint growth technology. Through tuning the sampling periods of the sampled Bragg gratings (SBGs), frequency detuning between these two sections can be achieved. When the lasing wavelength is located on the falling edge of the reflection spectrum of the GR section, the detuned-loading effect will be introduced, which can greatly enhance the modulation bandwidth.

When a π phase shift is introduced into the middle of the GR section, two steep falling edges will be formed around the center wavelength, such that the detuned-loading effect can be enhanced and achieved twice during the wavelength tuning. Furthermore, the introduction of a phase shift can effectively shorten the length of the chip due to the enhancement of the detuned-loading effect. Owing to its unique dual-falling-edges structure, the bandwidth can be improved even when the lasing wavelength shifts beyond the left falling edge due to a thermal effect in the actual test, in which condition the detuned-loading effect can be used twice, which greatly improves the yield. The performance is much better when the lasing wavelength is located on the left falling edge. In other words, a steeper falling slope can provide the enhanced detuned-loading effect on the improvement of modulation bandwidth and a better working characteristic for TS-DFB lasers.

Based on the detuned-loading effect, the modulation bandwidth of the TS-DFB laser is increased from 17.5 GHz for a single DFB laser to around 24 GHz when the lasing wavelength is located on the left falling edge, and it can be increased to 22 GHz when the lasing wavelength is located on the right side. Clear eye diagrams can be observed when the laser is modulated by 25 Gb/s and 30 Gb/s non-return-to-zero (NRZ) signals. An eight-channel laser array with precise wavelength spacing is investigated, with the SMSR > 36 dB. For lasers with no phase shift in the GR section, the lasing wavelength can easily fall outside the falling edge of the reflection spectrum during the tuning, in which case the detuned-loading effect will not work. Moreover, its falling edge is smoother and the maximum modulation bandwidth is only 22 GHz. The modulation characteristic of the TS-DFB laser with a uniform GR section is far inferior to the laser with a phase-shifted GR section.

## 2. Principle and Design

### 2.1. Principle of the REC Technique

In order to meet the growing demand for laser performance, many gratings with sophisticated structures have been applied. The REC technique using a sampled grating is proposed, where only conventional holographic exposure and the micrometer-level photolithography technique are required in the fabrication. In this method, complex grating structures can be quickly and precisely achieved by simply covering the mask above the seed grating and proceed with photolithography. The principle will be presented briefly in the following.

For the SBGs, the index modulation change, ∆n(z), can be described by:(1)$$∆n(z)=\frac{1}{2}s\left(z\right)\exp\left(j\frac{2\pi z}{\Lambda }\right)+c.c.$$
where  s(z)  is the periodic function of a sampling modulation and Λ is the period of the seed grating. According to the Fourier series expansion, s(z) can be written as:(2)$$s\left(z\right)=\underset{m}{\sum }F_{m}\exp\left(j\frac{2m\pi z}{P}\right)$$
where *P* is the sampling period and $F_{m}$ is the Fourier coefficient corresponding to the m th-order channel of the SBG. Substituting (2) into (1), we obtain:(3)$$∆n\left(z\right)=\underset{m}{\sum }\frac{1}{2}F_{m}\exp\left[j\left(\frac{2\pi z}{\Lambda }+\frac{2m\pi z}{P}\right)\right]+c.c.$$

From (3), we find that the SBG is actually a superposition of many subgratings with different grating periods. The m th equivalent grating period is given by:(4)$$\Lambda_{m}=\frac{P\Lambda }{P+m\Lambda }\;$$

When introducing a sampling period change,  ∆P , to  s(z) at ${\;z}_{0}$, the index modulation of the m th-order subgrating will be changed as:(5)$$∆n_{m}(z)=\left\{\begin{array}{c} F_{m}\exp\left(j\frac{2\pi z}{\Lambda }+j\frac{2m\pi z}{P}\right)+c.c.\;z\;\leq {\;z}_{0}\\ F_{m}\exp\left(j\frac{2\pi z}{\Lambda }+j\frac{2m\pi z}{P}-j\theta \right)+c.c{.\;z\;>\;z}_{0} \end{array}\right.$$
where the phase, θ, is obtained as: (6)$$\theta =2m\pi \frac{∆P}{P}\;$$

If *m* ≠ 0, an equivalent phase shift (EPS) can be achieved. According to (4) and (5), we can see that the SBG has multiple channels. The lasing wavelength and the spectrum spacing of each channel are mainly determined by the sampling period, *P*. In the practical design, by choosing a proper *P*, we can ensure that only the +1st-order channel is located within the wavelength range where the material has sufficient gain and all other channels are located outside the gain region. By changing the sampling period at equal intervals, a laser array can be fabricated.

### 2.2. Principle of the Detuned-Loading Effect

The principle of the detuned-loading effect of the DML is illustrated as follows. This effect occurs on the falling edge of the Bragg reflector mirror of DBR lasers (distributed Bragg reflector lasers) or DR lasers (distributed reflector lasers). In Figure 1a, ‘0’ represents the lasing wavelength position at a low injection current and ‘1’ represents the lasing wavelength position at a high injection current. In Figure 1b, as the gain section is modulated, the detuning between two sections sets the main mode on the long wavelength flank of the Bragg peak of the grating reflector, while the frequency up-chirp of the TS-DFB laser, due to changes in the refractive index under direct modulation, will shift the main mode closer to the Bragg peak of the GR section, in which condition the chirp of DMLs is translated into dynamic changes in the penetration depth and the loss of the DBR mirror. When chirp pushes the lasing wavelength to a shorter wavelength where the mirror has a higher reflection, mirror loss is reduced. Reduction in loss will increase the effective differential gain, and this can enhance the speed of DMLs beyond the limit of the material properties. 

### 2.3. Principle of the Simulation

The basic mode used in this paper is the time-domain dynamic model (TDDM) [21,22,23] and the schematic is shown in Figure 2. In this model, the laser cavity is uniformly divided into several subsections. During the process of light transmission, each subsection has its own independent parameters, such as grating coupling intensity, carrier density, photon density, gain, loss, refractive index, etc. If the division is detailed enough, that is, if the length of the microelement is small enough, each subsection can be considered uniform. According to the coupling wave theory, the optic field within the laser can be seen as a superposition of the forward wave, *F*(*z*, *t*), and the backward wave, *R*(*z*, *t*):(7)$${E(x,\;y,\;z,\;t)=\varphi (x,\;y)[\;F(z,\;t)e}^{{-i\beta }_{0}z}{+R(z,\;t)e}^{{i\beta }_{0}z}{\;]e}^{{i\omega }_{0}t}$$
where $\omega_{0}$ is the reference frequency corresponding to the Bragg wavelength, $\beta_{0}$ is the propagation constant at the Bragg wavelength and φ(x, y) is the distribution function of the mode in the waveguide.

The coupled wave equations of the forward wave, *F*(*z*, *t*), and the backward wave, *R*(*z*, *t*), are derived from Maxwell’s equations and can be written as:(8)$$\frac{1}{c_{g}}\frac{\mathrm{dF}\left(z,\;t\right)}{\mathrm{dt}}+\frac{\mathrm{dF}\left(z,\;t\right)}{\mathrm{dz}}=(G-i\delta )F\left(z,\;t\right)+\mathrm{ikR}\left(z,\;t\right){+S}_{f\;}\left(z,\;t\right)\;\frac{1}{c_{g}}\frac{\mathrm{dR}\left(z,\;t\right)}{\mathrm{dt}}-\frac{\mathrm{dR}\left(z,\;t\right)}{\mathrm{dz}}=(G-i\delta )R\left(z,\;t\right){+\mathrm{ik}}^{*}F\left(z\;,t\right){+S}_{r}\left(z,\;t\right)$$
where k is the coupling coefficient of the grating in the waveguide, describing the coupling of the forward and backward waves in the grating; G and δ are the mode gain and detuning factor, respectively; and $S_{f\;}\left(z,\;t\right)$ and $S_{r}\left(z,\;t\right)$ characterize the spontaneous emission noise within the waveguide, providing incentives for oscillation. When light is transmitted, the gain obtained can be written as:(9)$$G(z,\;t)=\frac{\Gamma \;g\;\ln(\frac{N(z,\;t)}{N_{0}})}{2(1+\varepsilon P(z,\;t))}-\frac{\alpha (z,\;t)}{2}$$
where g is the gain coefficient, $N_{0}$ is the transparency carrier density, α(z, t) is the internal loss caused by waveguide scattering and quantum-wells absorption, and P(z, t) represents the photon density within the cavity, which can be written as:(10)$${P(z,\;t)=|F(z,\;t)|}^{2}{+|R(z,\;t)|}^{2}$$

The parameter δ in Equation (8) is the detuning factor due to the change in the refractive index in the waveguide, representing the degree to which the lasing center wavelength deviates from the Bragg condition, and can be written as:(11)$$\delta =\frac{\omega_{0}}{c}{(n}_{\mathrm{eff},0}+∆n)-\frac{\pi }{\Lambda }$$
where $n_{\mathrm{eff},0}$ is the effective refractive index at transparency carrier density, Λ is the period of seed grating and ∆n is the is the refractive index change caused by the current injection, which is given by:(12)$$∆n=-\frac{\lambda_{0}}{4\pi }{\Gamma \;\alpha }_{m}\;\ln(\frac{N(z,\;t)}{N_{0}})$$
where ${\;\alpha }_{m}$ is the linewidth enhancement factor.

When current is injected into the laser, the internal carrier and photon density will change. Substituting these parameters into the time-domain carrier rate equation, the time-dependent carrier density rate equation changing with time will be obtained:(13)$$\frac{\mathrm{dN}}{\mathrm{dt}}=\frac{\eta_{i}J}{{\mathrm{ed}}_{\mathrm{act}}}-\;\mathrm{AN}\;-{\;\mathrm{BN}}^{2}-{\;\mathrm{CN}}^{3}-\frac{c_{g}\;g\;\ln(\frac{N(z,\;t)}{N_{0}})}{\left(1+\varepsilon P(z,\;t)\right)}$$
where J is the current density, $d_{\mathrm{act}}$ is the thickness of active layer, A is the linear recombination coefficient, B is the spontaneous recombination coefficient and C is the Auger recombination coefficient.

The boundary conditions of the forward light field and the backward light field at the interface can be written as:(14)$${F(z=0)=r}_{L}R(z=0)\;{R(z=L)=r}_{R}F(z=L)$$
where $r_{L}$ and $r_{R}$ are the reflectivity of the front and rear surfaces of the laser, respectively. Then, the output power of the laser at the surface can be written as:(15)$$P_{\mathrm{front}}{=h\nu (\mathrm{wd}}_{\mathrm{act}}v_{g}/\Gamma )|(1-r_{R}{)F(L,t)|}^{2}\;P_{\mathrm{back}}{=h\nu (\mathrm{wd}}_{\mathrm{act}}v_{g}/\Gamma )|(1-{\;r}_{L}{)R(0,t)|}^{2}$$
where h is Planck’s constant, ν is the reference frequency and w is the waveguide width.

### 2.4. The Design of the TS-DFB Laser

The epitaxial structure of the chip under production is shown in Figure 3a, where traditional metal–organic chemical vapor deposition (MOCVD) growth is used. During the first epitaxial growth, the N-InP buffer, the lower separate-confinement-heterostructure (SCH) layer, the AlGaInAs muti-quantum-wells (MQW) active region, and the upper SCH layer and the grating layer are grown on the N-InP substrate in turn. The SBG is achieved by holographic exposure combined with photolithography. Then, the P-InP cladding layer, the P-InGaAsP etch stop layer and the P-InP ridge waveguide (RWG) are fabricated. Finally, in order to suppress the Fabry–Perot modes of the lasers, AR coatings with reflectivity less than 1% are deposited on both facets.

A schematic of the proposed laser structure is depicted in Figure 3b. The laser is composed of two sections, which can be seen as an active DFB section and a grating reflection section. For convenience, we will refer to these two regions as section Ⅰ and section Ⅱ in the following parts. These two sections can be injected into the different currents, I_1_ and I_2_, respectively. These two sections share the same seed grating period but have slightly different sampling periods. By setting different sampling periods in each section, wavelength detuning with high accuracy can be achieved. There will be no current crosstalk phenomenon due to electrical isolation added in the middle of these two sections. 

A structure schematic of the SBG in the TS-DFB laser is shown in Figure 4a. The sampling structure is formed by a conventional holographic exposure combined with photolithography. In our design, section Ⅰ works as a lasing section where an abrupt π phase shift is introduced in the middle of the grating structure to guarantee a single-longitudinal-mode (SLM) operation. Section Ⅱ works as a grating reflector to provide section I with high reflectivity, but no laser oscillation will happen in this section. A π phase shift is introduced in the middle of section Ⅱ to provide a sharp falling slope and an extra falling edge. The phase is continuous across the interface between the two adjacent sections, and no additional phase shift is introduced. Both facets are anti-reflection (AR) coated to avoid the influence of the random phase. The transfer matrix method (TMM) [23,24] and the time-domain dynamic model (TDDM) are utilized for simulation. The passive spectra of reflection and transmission are simulated by the TMM model, and the small-signal curve and the eye diagram are simulated based on the TDDM model. The parameters used in the simulation are listed in Table 1. Some of these parameters are derived from previously fabricated epitaxial wafers and others are obtained from reference [21], such as the bimolecular recombination coefficient and the Auger recombination coefficient.

Figure 4b plots the transmission spectrum of section Ⅰ and the reflection spectrum of section Ⅱ. Due to the introduction of a π phase shift in the center of section Ⅱ, we can clearly observe a zero point in the middle of the reflection spectrum and there will be two falling edges. Such that when the lasing mode is located on any falling edge, the detuned-loading effect can be achieved, which improves the yield of the chip. Moreover, due to the introduction of the π phase shift in section Ⅱ, the width of the falling edge is narrowed and the slope is higher.

## 3. Simulation Results

To investigate the enhancement of the detuned-loading effect in this structure and prevent the grating reflector from emitting light due to excessive current injection, the current injected into section Ⅱ was set to 0 mA. By changing the sampling period to place the lasing wavelength at different positions of the falling flank, the variation in its modulation bandwidth was studied. We divided the left and right falling edges of the reflection spectrum into four parts spaced by their width and named their respective endpoints positions A, B, C, D and E, as shown in Figure 5, to study the working characteristics when lasing wavelengths fall at different positions.

Figure 6a,b plot the small-signal modulation response curves when the lasing wavelength is located at corresponding positions of the left and right falling edges when the injected current in section Ⅰ is 100 mA. The black dashed line represents that the value of the small-signal modulation response is −3 dB. As the penetration depth into the grating reflector increases, the reflection spectrum becomes steeper and the detuned-loading effect is enhanced. The modulation bandwidth increases from 17.8 GHz to 24 GHz, and the relaxation oscillation frequency can be enhanced from 8.6 GHz to 13 GHz. When the lasing wavelength is located on the right side, the modulation bandwidth increases from 17.6 GHz to 22 GHz, and the relaxation oscillation frequency varies from 8.4 GHz to 11 GHz.

It can be seen that in our structure the detuned-loading effect can work when the lasing wavelength is located on both falling edges of the reflection spectrum, which is a prominent advantage over those TS-DFB lasers with uniform GR sections. Owing to its unique dual-falling-edges structure, the lasing wavelength is difficult to drift out of the reflection spectrum range due to thermal effects during actual testing. The left falling edge has a steeper slope, resulting in a larger relaxation oscillation frequency and modulation bandwidth at different positions compared with the corresponding results when the lasing wavelength is located on the right side.

In addition, we studied the output characteristics when the lasing wavelength is located at different positions of both falling edges. The result is shown in Figure 7. With the increase in the mirror loss of the grating reflector, the output power of the left facet decreases gradually. The output power varies from 42 mW to 37 mW for the left flank and decreases from 40.8 mW to 31.1 mW for the right flank when the injection current I_1_ is 100 mA. It can be found that the output power of the right side is smaller than that of the left side, and this can be explained as follows. Due to the introduction of the π phase shift in the middle of section Ⅱ, the left falling flank is steeper and the width is narrower than the right. Therefore, the reflectivity of the corresponding position on the right falling edge is smaller. 

Comparing Figure 6 and Figure 7, we can find that the 3 dB modulation bandwidth can be enhanced with the increase in the mirror loss, while the output power of the facet is reduced. From the results for the simulated modulation bandwidth and the output power, the performance is much better when the lasing wavelength is located at the left falling edge. In other words, a steeper falling slope can provide the enhanced detuned-loading effect on the improvement of the modulation bandwidth and a better working characteristic for TS-DFB lasers.

The comparative analyses of the simulation results when the lasing wavelength falls on both falling flanks are shown in Figure 8. It can be seen that as the detuning between the lasing wavelength and the Bragg reflection peak increases, the modulation bandwidth gradually increases and the output power continuously decreases. At the same time, when the lasing wavelength falls on the left falling flank with a higher slope efficiency, the improvement in the modulation bandwidth is higher and the result is much superior to the right side. The steeper slope on the left side contributes to greater change in the reflectivity difference of the Bragg reflector under equal blue-shift chirp caused by current injection, resulting in a stronger detuned-loading effect. When the lasing wavelength falls at position A on both sides of the Bragg reflection spectrum, the results of the simulated relaxation oscillation frequency and modulation bandwidth are similar to the one-section DFB laser (OS-DFB), where its modulation characteristics have not been effectively improved and have even deteriorated. This is because the thermal effect was not taken into account during the simulation; current injection can only reduce the effective refractive index of the material, and thus the lasing wavelength will blue-shift to the rising edge of the reflection spectrum, in which case the detuned-loading effect does not work.

The simulated lasing spectrum is shown in Figure 9. The TS-DFB laser maintains SLM operation, and the side-mode suppression ratio is larger than 42 dB due to the introduction of the π phase shift in the middle of section Ⅰ.

For comparison, the response characteristic of the conventional OS-DFB laser with a cavity length of 400 μm is also given in Figure 10a. When the bias current of the active DFB laser is 100 mA, the 3 dB modulation bandwidth of the OS-DFB laser is only about 17.5 GHz, while the maximum 3 dB modulation bandwidth of the TS-DFB laser is 24 GHz and 22 GHz when the lasing wavelength is located on the left and right falling flank, respectively, where the increase of 6.5 GHz in the modulation bandwidth has been achieved. Figure 10b plots the simulated light–current characteristics of the TS-DFB laser and the OS-DFB laser. Apparently, the incorporation of the grating reflector can greatly improve the working performance of the laser, characterized by reducing threshold current and improving the output efficiency. The threshold current of the TS-DFB laser reduces to 17 mA. When the injection current I_1_ is 100 mA, the maximum output power is 42 mW and the slope efficiency is increased to 0.506 mW/mA. Compared with the conventional OS-DFB laser, the threshold current is decreased by 7 mA and the output power is increased by 18 mW.

Current injection increases carrier density, and this will decrease the refractive index of the material. The transmission spectrum of the DFB section will be blue-shifted with a decreased refractive index when the current is injected into section Ⅰ. Thus, when the lasing wavelength is designed to locate at the position with the lowest reflectivity, the actual wavelength will drift to the position with higher reflectivity. This can explain why the output power of the laser at position E is still higher than that of the conventional OS-DFB laser. 

When the lasing wavelength is located at position D, where the modulation bandwidth is around 22 GHz, NRZ modulation is performed. As shown in Figure 11, the modulation rates are 25 Gb/s, 30 Gb/s, 35 Gb/s and 40 Gb/s. The bias current is set to 100 mA and the modulation amplitude is 100 mA. Clearly, the eyes of the 25 Gb/s and 30 Gb/s are all well opened. However, due to the limitation of the modulation bandwidth, the result of the 40 Gb/s eye diagram is not ideal. In our simulation, the current injected into section Ⅱ was 0 mA, so the opening of the eye diagram was relatively limited. But according to previous testing experience, the opening would be greatly improved after current injection in section Ⅱ.

According to the REC theory discussed above, an eight-channel TS-DFB laser array with a wavelength around 1550 nm has been investigated. The simulated spectra of the laser array when the injection current, I_1_, is 100 mA is presented in Figure 12a. The lasing wavelength of the eight channels varies from 1547.7 nm to 1553.45 nm, and the minimum SMSR of the lasers is 36.21 dB. A good SLM operation can be achieved for all of the eight-channel lasers. The zeroth Bragg wavelength is set to 1645 nm, which is far away from the gain spectrum, and no zeroth lasing will be observed in the spectrum for any injection current, which ensures that only the +1st sampling wavelength oscillates in the laser. In Figure 12b, we show a linear fitting of the lasing wavelength. The slope efficiency of the fitting curve is 0.783 nm/count. The wavelength spacing of the laser array is designed as 0.8 nm, which is close to the simulation result. The wavelength residual of each laser channel can also be calculated after the linear fitting in Figure 12c. According to the calculation, the minimum wavelength residual is 0.035 nm.

In addition, the effect of the detuned-loading effect on the modulation bandwidth enhancement in the TS-DFB lasers with uniform GR sections has been investigated. The parameters used in the simulation were exactly the same as those listed in Table 1, and the only difference between these two structures is that there is no phase shift introduced in the grating reflector. As can be seen from the reflection spectrum in Figure 13, the slope of the falling edge is smoother.

The small-signal modulation response curves for the 100 mA current injected in section Ⅰ when the lasing wavelength is located at different positions of the falling flank is given in Figure 14a. With the increase in the mirror loss of the grating reflector, the 3 dB modulation bandwidth can be increased from the initial 17.5 GHz to 22 GHz, i.e., the modulation bandwidth is increased by 4.5 GHz. However, when comparing this structure with the one previously mentioned, we find that the bandwidth improvement of the detuned-loading effect in the TS-DFB lasers with uniform grating reflectors is far inferior to the laser with a π phase shift in the grating reflector. Moreover, in this structure, the detuned-loading effect will not act when the lasing wavelength shifts outside the falling edge of the reflection spectrum. For lasers with a phase shift in the grating reflector, the detuned-loading effect is not only stronger, but also, owing to its unique dual-falling-edges structure, the bandwidth can be improved even when the lasing wavelength shifts beyond the left falling edge due to thermal effects. 

## 4. Conclusions

A novel high-speed directly modulated TS-DFB semiconductor laser based on the REC technique is proposed and investigated. A π phase shift is introduced into the reflection grating which can provide a narrow-band reflection region with sharp falling slopes on both sides of the reflection spectrum, such that the detuned-loading effect can be enhanced. The modulation bandwidth is increased from 17.5 GHz for the single DFB laser to 24 GHz when the lasing wavelength is located on the left falling edge of the TS-DFB laser and can be increased to 22 GHz for the right falling edge based on the detuned-loading effect. By performing NRZ modulation on the laser, we obtained clear eye diagrams at 25 Gb/s and 30 Gb/s. An eight-channel TS-DFB laser array at a wavelength around 1550 nm has been investigated. The lasing wavelength of the eight channels varies from 1547.7 nm to 1553.45 nm, and the maximum SMSR of the lasers is 43 dB. According to the result of linear fitting, a minimum wavelength residual of 0.035 nm was achieved. Moreover, we have investigated the performance of TS-DFB lasers with uniform grating reflectors for comparison. We find that the bandwidth improvement of the detuned-loading effect in the TS-DFB lasers with uniform grating reflectors is far inferior to those with a π phase shift in the grating reflector. In addition, precise wavelength control is required to prevent the lasing wavelength shifting outside the falling flank of the reflection spectrum due to thermal effects, which greatly increases the difficulty of fabrication.

The structure proposed has great reference value for fabricating high-speed modulated DFB semiconductor lasers and can provide inspiration for future chip designs.

## Figures and Tables

**Figure 1 micromachines-14-01994-f001:**
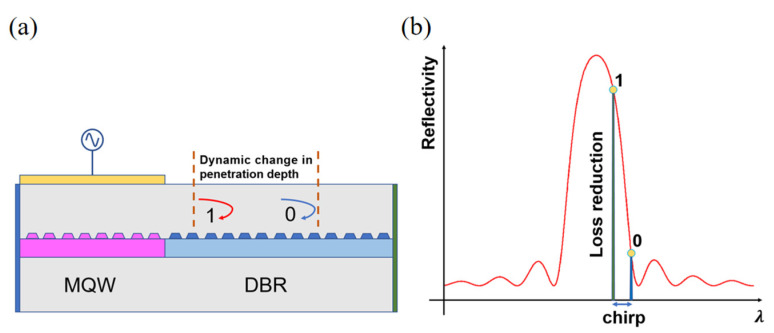
(**a**) Structure of the DR laser; (**b**) Dynamic change of the lasing wavelength under modulation.

**Figure 2 micromachines-14-01994-f002:**
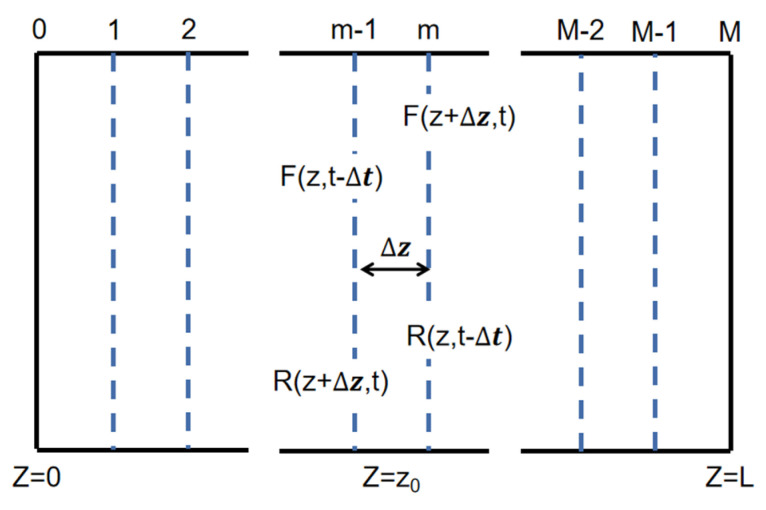
Schematic of the time-domain dynamic model.

**Figure 3 micromachines-14-01994-f003:**
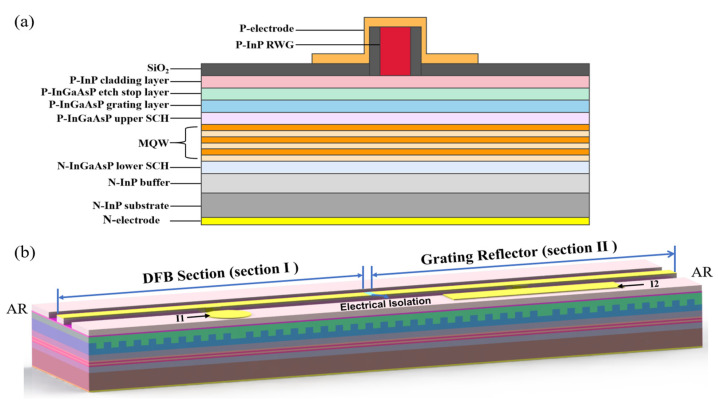
(**a**) Epitaxial structure of the chip under production. (**b**) Schematic of the TS-DFB laser.

**Figure 4 micromachines-14-01994-f004:**
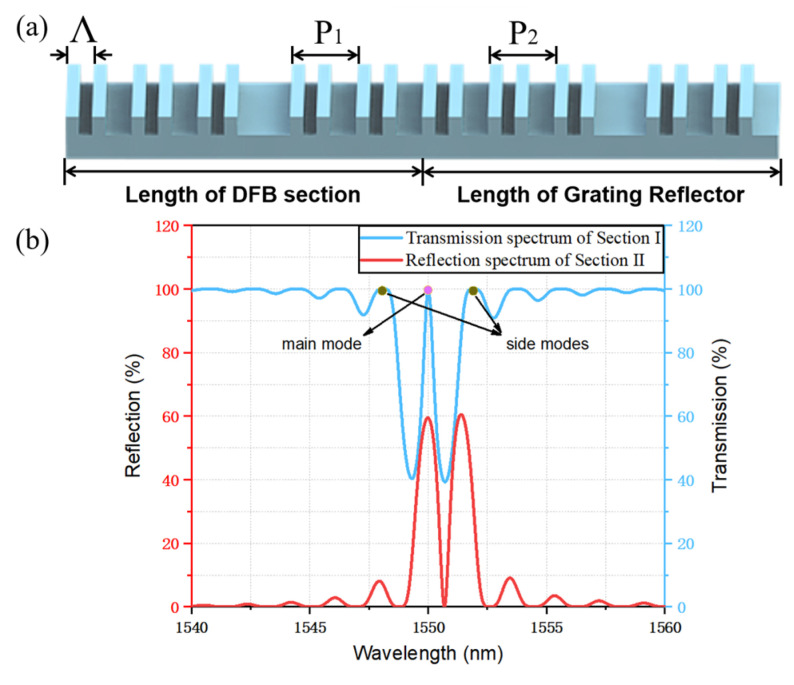
(**a**) Schematic of the SBG in the TS-DFB laser. (**b**) Calculated transmission and reflection spectra of the SBG in different sections.

**Figure 5 micromachines-14-01994-f005:**
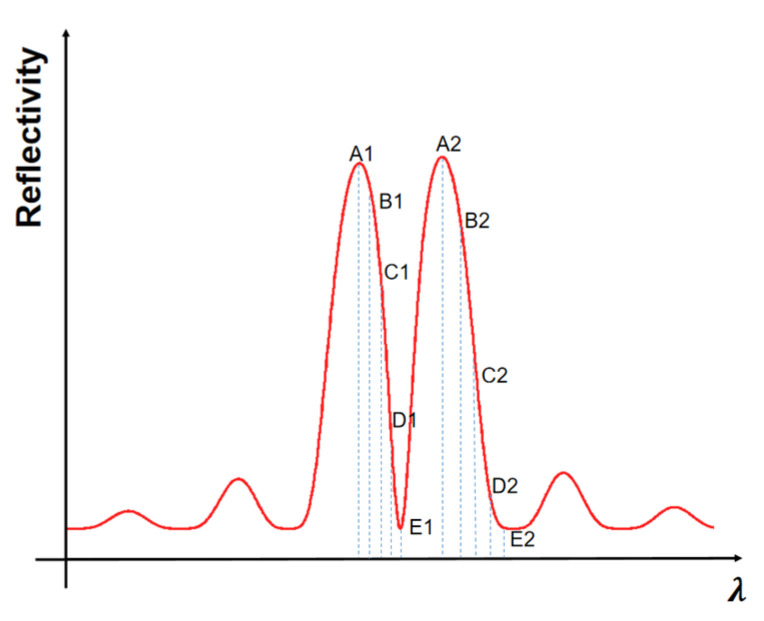
Schematic diagram of different operating wavelength positions.

**Figure 6 micromachines-14-01994-f006:**
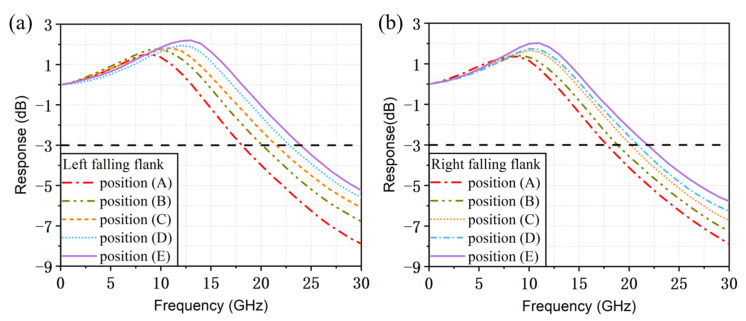
Small-signal intensity modulation response for different positions at (**a**) the left falling flank and (**b**) the right falling flank when I_1_ is 100 mA.

**Figure 7 micromachines-14-01994-f007:**
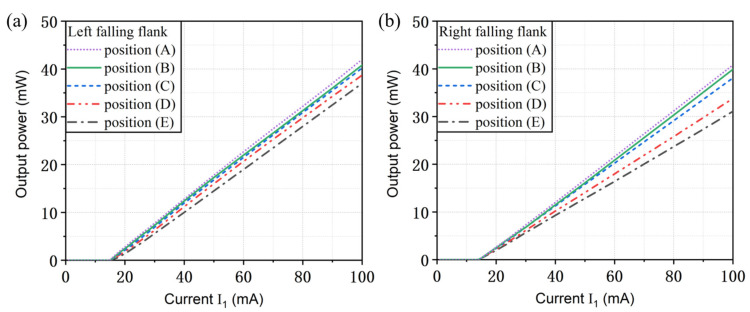
Light–current characteristics for different positions at (**a**) the left falling flank and (**b**) the right falling flank.

**Figure 8 micromachines-14-01994-f008:**
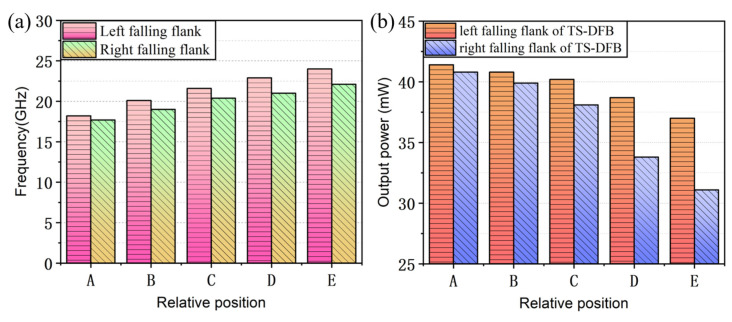
(**a**) Comparison of 3 dB modulation bandwidth between the left and right falling flank. (**b**) Comparison of facet output power between the left and right falling flank.

**Figure 9 micromachines-14-01994-f009:**
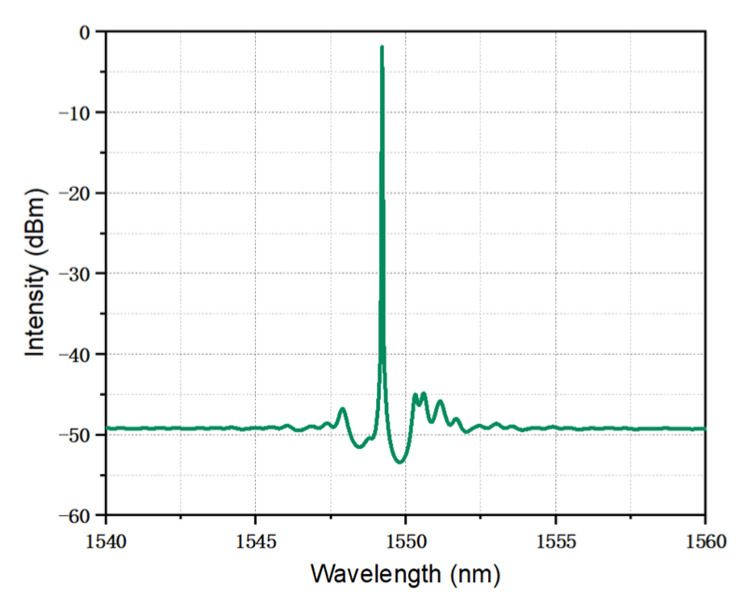
Lasing spectrum of the TS-DFB laser when the injection current is 100 mA.

**Figure 10 micromachines-14-01994-f010:**
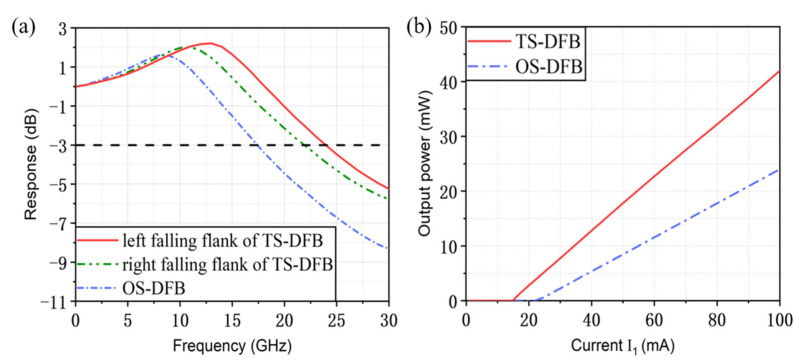
(**a**) Optimal response curves for the left and right falling edges of the TS-DFB laser and the OS-DFB laser when I_1_ is set to 100 mA. (**b**) Simulated light–current characteristics of the TS-DFB laser and the OS-DFB laser.

**Figure 11 micromachines-14-01994-f011:**
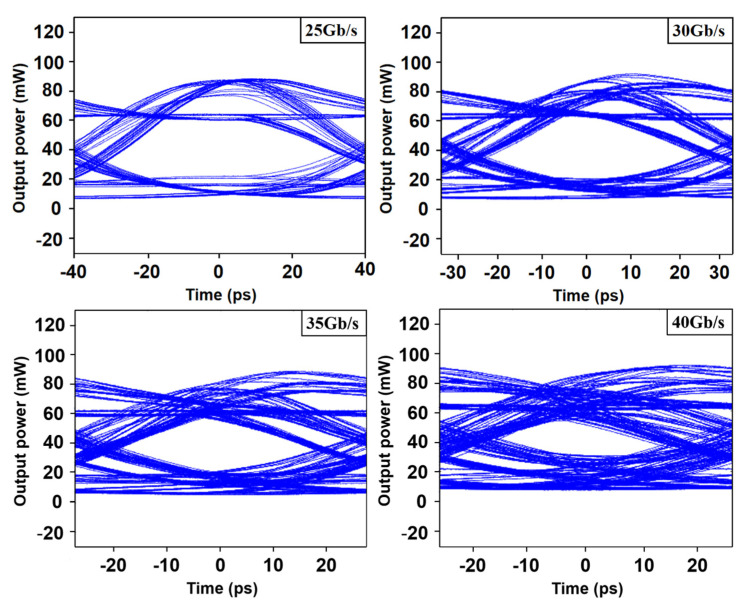
Eye diagrams of the laser modulated at 25 Gb/s, 30 Gb/s, 35 Gb/s and 40 Gb/s.

**Figure 12 micromachines-14-01994-f012:**
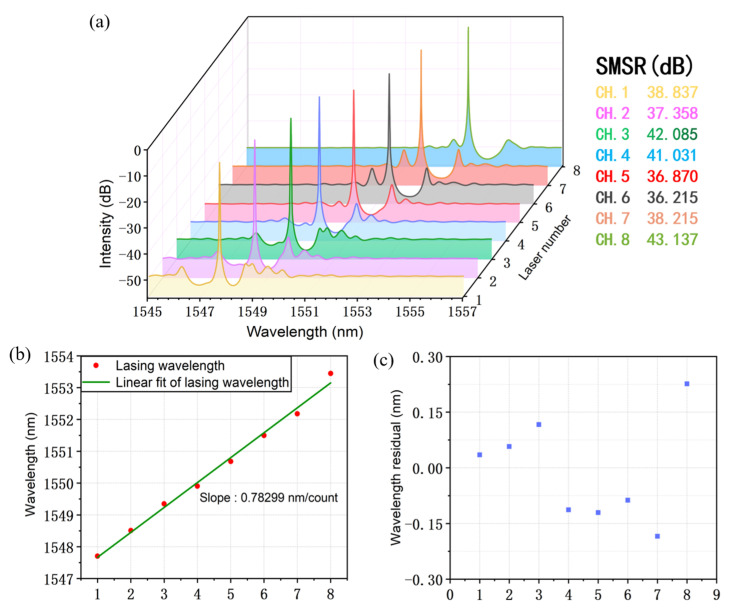
(**a**) Simulated lasing spectra of the eight-channel TS-DFB laser array when I_1_ is 100 mA. The table listing on the right shows the SMSR of each channel. (**b**) The corresponding lasing wavelengths of the laser array (red dots). The green line is the linear fitting line of the wavelengths. (**c**) Wavelength residuals after the linear fitting.

**Figure 13 micromachines-14-01994-f013:**
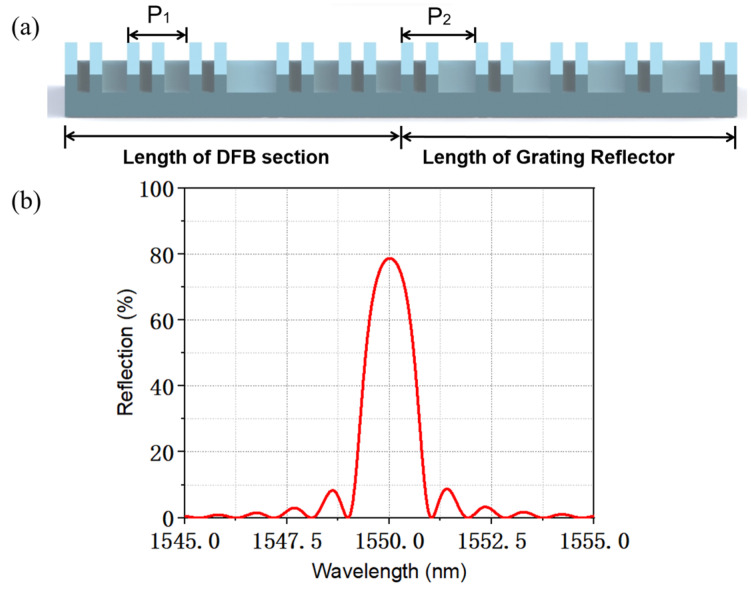
(**a**) Schematic of the SBG in the TS-DFB laser with uniform section Ⅱ. (**b**) Calculated reflection spectrum of section Ⅱ.

**Figure 14 micromachines-14-01994-f014:**
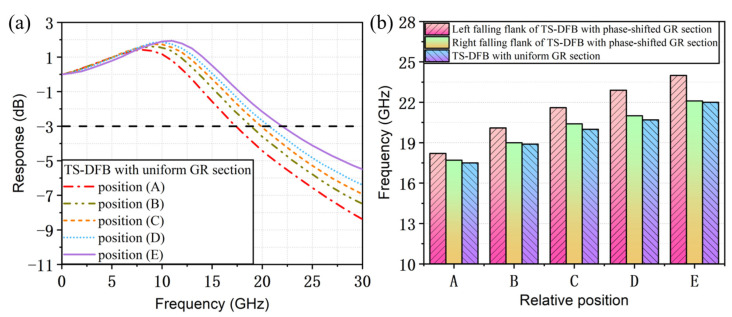
(**a**) Small-signal intensity modulation response for different positions at the falling flank. (**b**) Comparison of 3 dB modulation bandwidths between the TS-DFB with and without a phase shift in the GR section.

**Table 1 micromachines-14-01994-t001:** Parameters used in the simulation.

Parameter	Symbol	Value
Seed grating period (nm)	Λ0	256.71
Sampling period of DFB section (μm)	P1	4.189
Length of DFB section (μm)	L1	400
Length of grating reflector (μm)	L2	400
Active layer width (μm)	Wa	2
Active layer thickness (μm)	da	48
Effective refractive index	neff	3.204
Group refractive index	ng	3.6
Linewidth enhancement factor	αh	2
Gain coefficient (cm^−1^)	g	1100
Internal loss (cm^−1^)	α	10
Optical confinement factor	Γ	0.1
Monomolecular recombination coefficient (10^9^ s^−1^)	A	1
Bimolecular recombination coefficient (10^−10^ cm^3^ s^−1^)	B	1
Auger recombination coefficient (10^−29^ cm^6^ s^−1^)	C	7.5
Transparency carrier density (10^24^ m^−3^)	Ntr	1
Nonlinear gain saturation coefficient (10^−23^ m^−3^)	ε	4

## Data Availability

The data presented in this study are available on request from the corresponding author. The data are not publicly available due to confidentiality request.

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
