# Peer review of "Research on an Enhanced Detuned-Loading Effect in Integrated Two-Section DFB Lasers with High Modulation Bandwidths"

_micromachines, 2023, doi:10.3390/mi14111994_

Round 1

Reviewer 1 Report

In this manuscript, the authors used the Detuned-Loading Effect to enhance the bandwidth of the DFB lasers. The manuscript is well prepared. The method is scientifically correct, and the results and discussion can support the conclusion. However, the manuscript is lack of novelty. In my opinion, the manuscript can be published in this journal after the major correction.

I have several comments on the manuscript.

1. The seed grating period in Table 1 is 256.71 nm, how this parameter can be achieved in the processing.

2. It would be better if the authors could add where the parameters listed in Table 1 come from.

3. It would be better if the authors could add the epitaxy structure of the lasers.

4. In Fig. 10, the bit rate is not large for the lasers with over 20GHz bandwidth, what is the reason for this, and how the bit rate can be increased.

5. Using Detuned-Loading Effect and two-section DFB structure is not new, it would be better if the authors could explain the novelty and the significance of the manuscript.

6. There are some typos in the manuscript.

No

Author Response

Response to reviewers’ comments

Thank the reviewers and the associate editor for these constructive comments concerning my manuscript entitled “Research on Enhanced Detuned-Loading Effect in Integrated Two-Section DFB Lasers with High Modulation Bandwidth”.

These comments are all valuable and very helpful for improving my paper. We have studied the comments and revised our manuscript accordingly. These amendments have also been marked using red color in the revised manuscript. Our responses to the reviewers’ comments are listed as follows.

Review 1

In this manuscript, the authors used the Detuned-Loading Effect to enhance the bandwidth of the DFB lasers. The manuscript is well prepared. The method is scientifically correct, and the results and discussion can support the conclusion. However, the manuscript is lack of novelty. In my opinion, the manuscript can be published in this journal after the major correction:

  1. The seed grating period in Table 1 is 256.71 nm, how this parameter can be achieved in the processing.

Answer:

The grating structure in our paper is designed by REC technique, so it actually contains many sub-channels. In order to obtain sufficient gain, we always choose +1st channel to lase. The lasing wavelength is set to 1550nm. To achieve SLM operation, the 0st bragg wavelength must be away from the gain spectrum and is chosen to be 1645nm.

According to the Bragg grating theory, the Bragg wavelength of the seed grating is

, where neff is the effective refractive index of the grating and Λ0 is the period of the grating. When submitting neff = 3.204 and λB=1645 nm into the equation, we can get that the seed grating period is 256.71 nm. During the fabrication, the seed grating is usually achieved by holographic exposure.

  1. It would be better if the authors could add where the parameters listed in Table 1 come from.

Answer:

Thank you for your suggestions. We will add the introduction in this paper.

Some of these parameters are derived from previously fabricated epitaxial wafers and others are  obtained from reference, such as Bimolecular recombination coefficient and Auger recombination coefficient.

  1. It would be better if the authors could add the epitaxy structure of the lasers.

Answer:

Thank you for your suggestions.

The fabrication processing of the epilayer structure is described as follows:

The epitaxial structure of the chip under production is shown in Figure 3. (a), where traditional metal-organic chemical vapor deposition (MOCVD) growth is used. During the first epitaxial growth, the N-InP buffer, lower separate-confinement-heterostructure (SCH) layer, AlGaInAs muti-quantum wells (MQW) active region, upper SCH layer and grating layer are grown on the N-InP substrate in turn. The SBG is achieved by holographic exposure combined with photolithography. Then, the P-InP cladding layer, P-InGaAsP etch stop layer and P-InP ridge waveguide (RWG) are fabricated. Finally, in order to suppress the Fabry-Perot modes of the lasers, AR coatings with reflectivity less than 1% are deposited on both facets.

  1. In Fig. 10, the bit rate isnot large for the lasers with over 20GHz bandwidth, what is the reason for this, and how the bit rate can be increased.

Answer:

The reason for this condition is described as follows:

When the lasing wavelength is located on position D, where the modulation bandwidth is around 22 GHz, NRZ modulation is performed. The modulation rates are 25 Gb/s, 30 Gb/s, 35 Gb/s, and 40 Gb/s, respectively. The bias current is set to 100 mA and the modulation amplitude is 100 mA. Clearly, the eyes of the 25 Gb/s and 30 Gb/s are all well opened. However, due to the limitation of modulation bandwidth the result of 40 Gb/s eye diagram is not ideal. In our simulation, the current injected into section â…¡ is 0 mA so the opening of the eye diagram is relatively limited. But according to previous testing experience, the opening would be greatly improved after current injection in section â…¡.

  1. Using Detuned-Loading Effect and two-section DFB structure is not new, it would be better if the authors could explain the novelty and the significance of the manuscript.

Answer:

Thank you for your suggestions.

The novelty and significance has been listed and organized as follows:

The grating structure is designed be REC technique, the production standard can be reduced to the micron level, which would greatly reduce the manufacturing cost.

Active DFB section and passive grating reflector share the same MQW and ridge structure, avoiding the butt-joint growth progress.

When a π phase-shift is introduced into the middle of GR section, a symmetrical hollow will appear in the center of the reflection spectrum. As that the detuned-loading effect can be achieved twice during the wavelength tuning. Furthermore, the introduction of phase shift can effectively shorten the length of chip due to the enhancement of detuned-loading effect. Owing to its unique dual-falling-edges structure the bandwidth can be improved even when the lasing wavelength shifts beyond the left falling edge due to thermal effect in actual test, which greatly improves the yield. the performance is much better when the lasing wavelength is located on the left falling edge. In other words, a steeper falling slope can provide the enhanced detuned-loading effect on the improvement of modulation bandwidth and a better working characteristic for TS-DFB lasers. Based on the detuned-loading effect, the modulation bandwidth of the TS-DFB laser is increased from 17.5 GHz for a single DFB laser to around 24 GHz when the lasing wavelength is located on the left falling edge and can be increased to 22 GHz when the lasing wavelength is located on the right side. Clear eye diagrams can be observed when the laser is modulated by 25 Gb/s and 30 Gb/s non-return-to-zero (NRZ) signal. An eight-channel laser array with precise wavelength spacing is investigated, with SMSR > 36 dB. For lasers with no phase-shift in the GR section, the lasing wavelength can easily fall outside the falling edge of the reflection spectrum during the tuning in which case the detuned-loading effect will not work. Moreover, its falling edge is smoother and the maximum modulation bandwidth is only 22 GHz. The modulation characteristic of the TS-DFB laser with uniform GR section is far inferior to the laser with a phase-shifted GR section.

  1. There are some typos in the manuscript.

Answer:

Thanks for your comments. 

We have reviewed the paper again and corrected the typos.

Thanks for your comments again.

Best wishes,

Chen xiangfei (On behalf of all authors)

Reviewer 2 Report

The authors theoretically study high-speed directly modulated two-section distributed feedback semiconductor laser. Using the proposed laser design, the authors show the possibility of increasing modulation bandwidth to around 24 GHz. This is important for developing optical network communication. The paper is well-organized and well written. The results are solid and promising. I have no suggestions for the authors how to improve their paper. So I'm recommend the paper for publication.

Author Response

Response to reviewers’ comments

Thank the reviewers and the associate editor for these constructive comments concerning my manuscript entitled “Research on Enhanced Detuned-Loading Effect in Integrated Two-Section DFB Lasers with High Modulation Bandwidth”.

These comments are all valuable and very helpful for improving my paper. We have studied the comments and revised our manuscript accordingly. These amendments have also been marked using red color in the revised manuscript. Our responses to the reviewers’ comments are listed as follows.

Review 2  

The authors theoretically study high-speed directly modulated two-section distributed feedback semiconductor laser. Using the proposed laser design, the authors show the possibility of increasing modulation bandwidth to around 24 GHz. This is important for developing optical network communication. The paper is well-organized and well written. The results are solid and promising. I have no suggestions for the authors how to improve their paper. So I'm recommend the paper for publication.

Answer: 

Thanks for your comments again.

Best wishes,

Chen xiangfei (On behalf of all authors)

Reviewer 3 Report

In the manuscript a novel high-speed directly modulated two-section distributed feedback (TS-DFB) semiconductor laser based on the detuned-loading effect is proposed and investigated. Grating structure is designed by reconstruction-equivalent-chirp (REC) technique.

Overall, I found the manuscript vague and lacking details on how the results were obtained. Especially:

1. The abstract does not clearly indicate that the research is of a design and simulation nature.

2. The novelty of the work could have been emphasized more clearly in the introduction.

3. In section 2.1, I got lost in the last sentence, where the quantity „p” appears for the first time.

4. Figure 1 in section 2.2 has two parts, but it is not explained what the second part shows.

5. Section 2.3 contains information about the method used, namely: „The transfer matrix method (TMM) is used for simulation. The parameters used in the simulation are summarized in Table 1.”

6. All other results in the manuscript also lack appropriate references to enable their reproduction.

In aims of the Micromachines journal one can find, among other things, that a research article must provide detailed information about the methods so that the results can be reproduced.

This manuscript does not meet this requirement and for this reason I cannot recommend it for publication in its current form.

Author Response

Response to reviewers’ comments

Thank the reviewers and the associate editor for these constructive comments concerning my manuscript entitled “Research on Enhanced Detuned-Loading Effect in Integrated Two-Section DFB Lasers with High Modulation Bandwidth”.

These comments are all valuable and very helpful for improving my paper. We have studied the comments and revised our manuscript accordingly. These amendments have also been marked using red color in the revised manuscript. Our responses to the reviewers’ comments are listed as follows.

Review 3 

In the manuscript a novel high-speed directly modulated two-section distributed feedback (TS-DFB) semiconductor laser based on the detuned-loading effect is proposed and investigated. Grating structure is designed by reconstruction-equivalent-chirp (REC) technique.

Overall, I found the manuscript vague and lacking details on how the results were obtained. Especially:

  1. The abstract does not clearly indicate that the research is of a design and simulation nature.

Answer: 

Thank you for your remarks on our work.

Now we have indicated in the abstract that this is a simulation study article and actually the designed chip is in production.

A novel high-speed directly modulated two-section distributed feedback (TS-DFB) semiconductor laser based on the detuned-loading effect is proposed and simulated.

  1. The novelty of the work could have been emphasized more clearly in the introduction.

Answer:

Thank you for your suggestions.

The novelty and significance  been listed and organized as follows:

  • The grating structure is designed be REC technique, the production standard can be reduced to the micron level, which would greatly reduce the manufacturing cost.
  • Active DFB section and passive grating reflector share the same MQW and ridge structure, avoiding the butt-joint growth progress.
  • When a πphase-shift is introduced into the middle of GR section, a symmetrical hollow will appear in the center of the reflection spectrum. As that the detuned-loading effect can be achieved twice during the wavelength tuning. Furthermore, the introduction of phase shift can effectively shorten the length of chip due to the enhancement of detuned-loading effect. Owing to its unique dual-falling-edges structure the bandwidth can be improved even when the lasing wavelength shifts beyond the left falling edge due to thermal effect in actual test, which greatly improves the yield. the performance is much better when the lasing wavelength is located on the left falling edge. In other words, a steeper falling slope can provide the enhanced detuned-loading effect on the improvement of modulation bandwidth and a better working characteristic for TS-DFB lasers. Based on the detuned-loading effect, the modulation bandwidth of the TS-DFB laser is increased from 17.5 GHz for a single DFB laser to around 24 GHz when the lasing wavelength is located on the left falling edge and can be increased to 22 GHz when the lasing wavelength is located on the right side. Clear eye diagrams can be observed when the laser is modulated by 25 Gb/s and 30 Gb/s non-return-to-zero (NRZ) signal. An eight-channel laser array with precise wavelength spacing is investigated, with SMSR > 36 dB. For lasers with no phase-shift in the GR section, the lasing wavelength can easily fall outside the falling edge of the reflection spectrum during the tuning in which case the detuned-loading effect will not work. Moreover, its falling edge is smoother and the maximum modulation bandwidth is only 22 GHz. The modulation characteristic of the TS-DFB laser with uniform GR section is far inferior to the laser with a phase-shifted GR section.

  1. In section 2.1, I got lost in the last sentence, where the quantity „p” appears for the first time.

Answer: 

Thank you for your remarks on our work.

I’m sorry this is a mistake and we've made changes. What we originally meant was:

By changing the sampling period at equal intervals, a laser array can be fabricated.

  1. Figure 1 in section 2.2 has two parts, but it is not explained what the second part shows.

Answer: 

Thank you for your remarks on our work.

The principle of detuned-loading effect of the DML is illustrated as follows. This effect occurs on the falling edge of the Bragg reflector mirror of DBR lasers (Distributed Bragg Reflector Lasers) or DR lasers (Distributed Reflector Lasers) in Figure 1. (a). ‘0’ represents the lasing wavelength position at low injection current ,’1’ represents the lasing wavelength position at high injection current. In Figure 1. (b),as the gain section is modulated, the detuning between two sections sets the main mode on the long wavelength flank of the Bragg peak of grating reflector, while the frequency up-chirp of the TS-DFB laser due to changes in refractive index under direct modulation will shift the main mode closer to the Bragg peak of GR section, in which condition the chirp of DMLs is translated into dynamic changes in the penetration depth and the loss of the DBR mirror. When chirp pushes the lasing wavelength to shorter wavelength where the mirror has a higher reflection, mirror loss is reduced. Reduction of loss will increase effective differential gain and this can enhance the speed of DMLs beyond the limit of the material properties.

  1. Section 2.3 contains information about the method used, namely: „The transfer matrix method (TMM) is used for simulation. The parameters used in the simulation are summarized in Table 1.”

Answer: 

Thank you for your remarks on our work.

The basic models used in our study are the transfer matrix method (TMM) and time-domain dynamic model (TDDM). The TMM is a conventional method and the principle of TDDM has been added in our paper.

  1. All other results in the manuscript also lack appropriate references to enable their reproduction.

Answer: 

Thank you for your suggestions.

The basic models used in our paper are the transfer matrix method (TMM) and time-domain dynamic model (TDDM). The passive spectra of reflection and transmission are simulated by TMM model, the small signal curve and eye diagram are simulated based on the TDDM model. The references for these two models have been added in our paper.

Thanks for your comments again.

Best wishes,

Chen xiangfei (On behalf of all authors)
